# An update on ozone profile trends for the period 2000 to 2016

Wolfgang Steinbrecht[1], Lucien Froidevaux[2], Ryan Fuller[2], Ray Wang[3], John Anderson[4], Chris Roth[5], Adam Bourassa[5], Doug Degenstein[5], Robert Damadeo[6], Joe Zawodny[6], Stacey Frith[7,8], Richard McPeters[7], Pawan Bhartia[7], Jeannette Wild[9,10], Craig Long[9], Sean Davis[11,12], Karen Rosenlof[11], Viktoria Sofieva[13], Kaley Walker[14], Nabiz Rahpoe[15], Alexei Rozanov[15], Mark Weber[15], Alexandra Laeng[16], Thomas von Clarmann[16], Gabriele Stiller[16], Natalya Kramarova[7,8], Sophie Godin-Beekmann[17], Thierry Leblanc[18], Richard Querel[19], Daan Swart[20], Ian Boyd[21], Klemens Hocke[22], Niklaus Kämpfer[22], Eliane Maillard Barras[23], Lorena Moreira[22], Gerald Nedoluha[24], Corinne Vigouroux[25], Thomas Blumenstock[16], Matthias Schneider[16], Omaira García[26], Nicholas Jones[27], Emmanuel Mahieu[28], Dan Smale[19], Michael Kotkamp[19], John Robinson[19], Irina Petropavlovskikh[29,12], Neil Harris[30], Birgit Hassler[31], Daan Hubert[25], and Fiona Tummon[32]

[1]Deutscher Wetterdienst, Hohenpeissenberg, Germany
[2]Jet Propulsion Laboratory, California Institute of Technology, Pasadena, CA, USA
[3]School of Earth and Atmospheric Sciences, Georgia Institute of Technology, Atlanta, GA, USA
[4]Department of Atmospheric and Planetary Sciences, Hampton University, Hampton, VA, USA
[5]Institute of Space and Atmospheric Studies, University of Saskatchewan, Saskatoon, Canada
[6]NASA Langley Research Center, Hampton, VA, USA
[7]NASA Goddard Space Flight Center, Silver Spring, MD, USA
[8]Science Systems and Applications Inc., Lanham, MD, USA
[9]NOAA/NWS/NCEP/Climate Prediction Center, College Park, MD, USA
[10]Innovim LLC, Greenbelt, MD, USA
[11]Chemical Sciences Division, NOAA ESRL, Boulder, CO, USA
[12]CIRES, University of Colorado, Boulder, CO, USA
[13]Finnish Meteorological Institute, Helsinki, Finland
[14]Department of Physics, University of Toronto, Canada
[15]Institute for Environmental Physics, University of Bremen, Germany
[16]Karlsruhe Institute of Technology, Institute of Meteorology and Climate Research, Karlsruhe, Germany
[17]Centre National de la Recherche Scientifique, Université de Versailles Saint-Quentin-en-Yvelines, Guyancourt, France
[18]Jet Propulsion Laboratory, California Institute of Technology, Wrightwood, CA, USA
[19]National Institute of Water and Atmospheric Research (NIWA), Lauder, New Zealand
[20]National Institute for Public Health and the Environment (RIVM), Bilthoven, The Netherlands
[21]BC Scientific Consulting LLC, Stony Brook, NY, USA
[22]Institute of Applied Physics and Oeschger Centre for Climate Change Research, University of Bern, Switzerland
[23]MeteoSwiss, Payerne, Switzerland
[24]Naval Research Laboratory, Washington, D.C., USA
[25]Royal Belgian Institute for Space Aeronomy (BIRA-IASB), Brussels, Belgium
[26]Izaña Atmospheric Research Centre (IARC), Agencia Estatal de Meteorología (AEMET), Santa Cruz de Tenerife, Spain
[27]School of Chemistry, University of Wollongong, Australia
[28]Institute of Astrophysics and Geophysics, University of Liège, Belgium
[29]Climate Monitoring Division, NOAA ESRL, Boulder, CO, USA
[30]Centre for Atmospheric Informatics and Emissions Technology, Cranfield University, United Kingdom
[31]Bodeker Scientific, Alexandra, New Zealand
[32]ETH Zürich, Switzerland

*Correspondence to:* Wolfgang Steinbrecht (wolfgang.steinbrecht@dwd.de)

**Abstract.** Ozone profile trends over the period 2000 to 2016 from several merged satellite ozone data sets and from ground-based data measured by four techniques at stations of the Network for the Detection of Atmospheric Composition Change indicate significant ozone increases in the upper stratosphere, between 35 and 48 km altitude (5 and 1 hPa). Near 2 hPa (42 km), ozone has been increasing by about 1.5% per decade in the tropics (20°S to 20°N), and by 2 to 2.5% per decade in the 35° to 60° latitude bands of both hemispheres. At levels below 35 km (5 hPa), 2000 to 2016 ozone trends are smaller and not statistically significant. The observed trend profiles are consistent with expectations from chemistry climate model simulations. This study confirms positive trends of upper stratospheric ozone already reported, e.g., in the WMO/UNEP Ozone Assessment 2014, or by Harris et al. (2015). Compared to those studies, three to four additional years of observations, updated and improved data sets with reduced drift, and the fact that nearly all individual data sets indicate ozone increase in the upper stratosphere, all give enhanced confidence. Uncertainties have been reduced, for example for the trend near 2 hPa in the 35° to 60° latitude bands from about ±5% ($2\sigma$) in Harris et al. (2015) to less than ±2% ($2\sigma$). Nevertheless, a thorough analysis of possible drifts and differences between various data sources is still required, as is a detailed attribution of the observed increases to declining ozone depleting substances and to stratospheric cooling. Ongoing quality observations from multiple independent platforms are key for verifying that recovery of the ozone layer continues as expected.

## 1 Introduction

Depletion of the stratospheric ozone layer by anthropogenic chlorine and bromine from ozone depleting substances (ODS) has been a world-wide concern since the 1970s (Stolarski and Cicerone, 1974; Molina and Rowland, 1974). Initially, studies predicted the largest ozone losses for the upper stratosphere, at about 42 km or 2 hPa (Crutzen, 1974). For the total column of ozone only moderate losses were predicted. Public perception of the situation changed dramatically with the discovery of the Antarctic ozone hole (Chubachi, 1984; Farman et al., 1985). The ozone hole is characterized by large ozone depletion throughout the lower stratosphere, which is due to heterogeneous reactions on the surface of Polar Stratospheric Clouds (Solomon, 1999). These important reactions had not been known and not been included in the early predictions. The large spring-time ozone losses over an entire continent were a huge surprise. The world's nations reacted to mounting evidence that ODS were harming the vital ozone layer, first by signing the International Vienna Convention for the Protection of the Ozone Layer in 1985, then by implementing the 1987 Montreal Protocol and its later amendments. Thanks to these agreements, the world-wide production and consumption of ODS have been eliminated almost completely since the early 1990s (WMO, 2007).

The Montreal Protocol has been very successful. The concentration of ODS in the atmosphere has been declining since the mid-1990s in the troposphere, and since the late 1990s also in the stratosphere (WMO, 2011). Scientific assessments of the state of the ozone layer have shown that the ozone layer is responding: The decline of ozone in the upper stratosphere stopped around 2000 (Newchurch et al., 2003; WMO, 2007). Total ozone columns have also stabilized (WMO, 2011, 2014). Given the current slow decline of ODS concentrations, we now expect ozone to increase accordingly in the stratosphere. However,

this small increase is not easily separated from concurrent variability and changes in temperature, atmospheric circulation, and solar ultraviolet flux (Jonsson et al., 2004; Reinsel et al., 2005; WMO, 2007, 2011).

The last WMO/UNEP ozone assessment (WMO, 2014), therefore, concluded that statistically significant increases of ozone had been observed only in the upper stratosphere (around 42 km or 2 hPa), but not at lower levels, and not for total ozone columns. About half of the increase in the upper stratosphere was attributed to declining ODS, the other half to declining temperature. This stratospheric cooling is caused by increasing $CO_2$ (Jonsson et al., 2004; Randel et al., 2016). Low temperature enhances ozone in the upper stratosphere, by slowing gas-phase destruction cycles and making ozone production more efficient.

Studies published after WMO (2014) have confirmed the tendency of ozone increasing in the upper stratosphere, but they also pointed out that instrument drifts and drift uncertainties might be larger than the 1 to 2% per decade assumed in WMO (2014). Hubert et al. (2016), for example, reported drifts and drift uncertainties between satellite and ground-based data exceeding 5% per decade for some instruments, and less than 2% per decade only for a few instruments. Harris et al. (2015) found larger differences between trends from some data sets, exceeding 6% per decade. Based on these larger differences, and the larger drift uncertainty estimates (Hubert et al., 2016), Harris et al. (2015) concluded that upward trends of upper stratospheric ozone might not be statistically different from zero.

The purpose of the present paper is to follow up on these studies, but with three to four more years of data, and with improved and additional data sets. Here we present initial results. A more comprehensive investigation of instrumental and merging uncertainties, and of uncertainties for different regression analyses is under way in the "Long-term Ozone Trends and Uncertainties in the Stratosphere" initiative (LOTUS), an activity of the Stratosphere-troposphere Processes And their Role in Climate project (SPARC) of the World Climate Research Programme (WCRP), see http://www.sparc-climate.org/activities/ozone-trends/.

## 2   Ozone Profile Data Records

The determination of ozone trends requires homogeneous data records that extend over several decades, because not only ozone variations associated with the quasi-biennial oscillation must be quantified well, but also the slow variations associated with the 11-year solar cycle (Newchurch et al., 2003; Steinbrecht et al., 2004). Available long-term records of ozone profile data start before 1990 and extend to the present (see also  Tegtmeier et al., 2013; Hassler et al., 2014; Tummon et al., 2015). Tables 1 and 2 summarize the merged satellite records and ground-based stations used in the present study.

### 2.1   Data Sources

The nadir viewing Solar Backscatter UltraViolet (SBUV) instruments on NASA and NOAA satellites have measured ozone profiles continuously since late 1978, covering the sunlit part of the globe, but with only coarse altitude resolution of 10 to 15 kilometers (McPeters et al., 2013). Orbit drifts, differences between individual instruments, instrument degradation, and some other problems require careful assessment, when generating a long-term data set from these measurements. Currently two SBUV based data sets (Version 8.60) are available: The merged SBUV MOD (release 6) ozone data set generated by NASA

(Frith et al., 2014), termed SBUV-NASA in the following, and the "coherent" SBUV data set generated by NOAA (Wild and Long, 2017), termed SBUV-NOAA in the following. The two data sets rely on the same SBUV instruments, but differ in the approach taken for merging their individual records (see also Frith et al., 2017).

Ozone profiles with higher vertical resolution (about 2 km), but also with sparser coverage, were provided by the satellite-borne Stratospheric Aerosol and Gas Experiments (SAGE I and SAGE II) and the Halogen Occultation Experiment (HALOE). These instruments measured in solar occultation geometry from 1979 to about 1982 (SAGE I), from late 1984 to 2005 (SAGE II), and from 1991 to 2005 (HALOE), see, e.g., Damadeo et al. (2013, 2014), and Remsberg (2008). Since 2002, the Optical Spectrograph and InfraRed Imaging System (OSIRIS) measures ozone profiles from ultraviolet light scattered in limb geometry (McLinden et al., 2012). SAGE II and OSIRIS ozone profiles have been combined by Bourassa et al. (2014) to produce a long-term data set, which has subsequently been improved by correcting for a tangent altitude drift of the OSIRIS instrument (Bourassa et al., 2017). Optionally, this data set also includes ozone profiles from the limb viewing instrument of the Ozone Mapping Profiler Suite (OMPS), which has operated since early 2012 (e.g., Flynn et al., 2014).

Using microwave emissions in limb geometry, the Microwave Limb Sounder (MLS) on the Aura satellite has been measuring many stratospheric trace gases since 2004, including ozone profiles with dense spatial sampling and a vertical resolution of 2.5 to 3 km in the stratosphere (Waters et al., 2006). SAGE, HALOE, and MLS ozone profiles have been combined in the Global OZone Chemistry And Related trace gas Data records for the Stratosphere (GOZCARDS, Froidevaux et al., 2015, newer version 2.20 used here) and in the Stratospheric Water and OzOne Satellite Homogenized data set (SWOOSH, Davis et al., 2016). GOZCARDS (v2.20) and SWOOSH (v2.6) are very similar (compare Fig. 1). Both rely to a large degree on the ozone records from SAGE II (1984 to 2005, version 7) and Aura-MLS (2004 to present, version 4.2). Both adjust ozone values from other satellites to those from SAGE II. GOZCARDS additionally uses SAGE I (version 5.9_rev) to extend the ozone record back to 1979.

For the period from August 2002 to April 2012, ozone profiles were also measured by the SCIAMACHY (= SCanning Imaging Absorption spectroMeter for Atmospheric CHartographY), GOMOS (= Global Ozone Monitoring by Occultation of Stars) and MIPAS (= Michelson Interferometer for Passive Atmospheric Sounding) instruments on board the European ENVISAT satellite. Positive ozone trends have been reported in the upper stratosphere for each of these instruments (Gebhardt et al., 2014; Kyrölä et al., 2013; Eckert et al., 2014). Unfortunately, ENVISAT failed in April 2012, and measurements ceased. The ESA Climate Change Initiative has generated a harmonized ozone profile data set (Sofieva et al., 2013; Rahpoe et al., 2015) from the ENVISAT instruments, the SMR (= Sub-Millimeter Radiometer) microwave instrument, the OSIRIS instrument, and ACE-FTS (= Atmospheric Chemistry Experiment Fourier Transform Spectrometer, see Bernath, 2017). This "ESA CCI" or "Ozone CCI" ozone profile record has recently been updated and extended, with SAGE II ozone profiles back to 1984, and with OMPS ozone profiles (2D retrieval from U. Saskatoon) from 2012 to the present (SAGE + CCI + OMPS, see Sofieva et al., 2017). Another new merged data set, following previous work by Laeng et al. (2017), combines the MIPAS (Fischer et al., 2008) ozone profile record (KIT/IMK processing) with the records from SAGE II and OMPS (NASA v2 retrieval). Because of short or lacking overlap periods, this SAGE + MIPAS + OMPS record relies on ACE-FTS as a transfer standard for matching

MIPAS high spectral resolution mode data (07/2002 until 03/2004) to MIPAS low spectral resolution mode data (01/2005 to 04/2012), and for matching the latter to OMPS data (after 02/2012).

While satellites provide near global coverage, the limited lifetimes of most satellite instruments makes the construction of consistent long-term records difficult, as indicated above. Long-term consistency, therefore, might be more easily achieved by ground-based measurements, albeit at the cost of only local coverage. Ground-based instruments have provided some of the longest available records for ozone trend analysis (e.g., Zanis et al., 2006; Nair et al., 2013). Therefore, the ground-based stations in Table 2 are used as an independent source for ozone trends in the present study. The longest ground-based ozone profile records for the upper stratosphere come from Dobson spectrometers operated in "Umkehr" mode (Petropavlovskikh et al., 2005, 2011). Umkehr ozone profiles have coarse altitude resolution, about 10 km. Long-term ground-based measurements of ozone in the upper stratosphere are also available from the Network for the Detection of Atmospheric Composition Change (NDACC, http://www.ndacc.org, Kurylo et al., 2016). These measurements started in the late 1980s and 1990s, using differential absorption lidars, microwave radiometers (Steinbrecht et al., 2009), and Fourier transform infrared spectrometers (FTIRs, Vigouroux et al., 2015). FTIR ozone profiles have coarse altitude resolution (8 to 15 km) and resolve only 3 layers in the stratosphere. Altitude resolution for the microwave radiometers is also 8 to 15 km. Lidars provide altitude resolution between 1 km (below 30 km) and 10 km (above 45 km).

A comprehensive intercomparison of limb-viewing satellite instruments with ground-based NDACC ozone sondes and lidars by Hubert et al. (2016) indicates that SAGE II and Aura-MLS, the primary instruments in many of the merged records, are very stable. If drifts exist, they are smaller than $\pm 2\%$ per decade in the 20 to 40 km region, and not statistically significant. Below 20 km and above 45 km uncertainties become larger, because of larger geophysical variation in the compared altitude ranges, and because of increasing measurement errors, see also Tegtmeier et al. (2013). Note that in Hubert et al. (2016), the OSIRIS V5.07 ozone data did exhibit a significant drift, up to 8% per decade near 40 km, also apparent in Harris et al. (2015). This drift has been corrected in the revised and updated OSIRIS V5.10 data set used here (Bourassa et al., 2017). Drifts of most SBUV instruments are less than 3 to 5% per decade, and are not statistically significant (Kramarova et al., 2013). Similarly, Rahpoe et al. (2015) report that drifts of several limb viewing instruments including ACE-FTS, MIPAS, and OSIRIS are typically less than 3% per decade (even for older processing versions), and not statistically significant. For the current study, newer data sets with reduced drifts were available (especially OSIRIS), and older data sets with apparent large drifts were not used (SAGE + GOMOS).

Table 3 compares data sets and trend periods used here, in WMO (2014), and in Harris et al. (2015). In addition to using three more years of data, the main difference between the present study and WMO (2014) is the use of four more satellite data sets: SBUV-NOAA, SWOOSH, SAGE + OSIRIS, and SAGE + MIPAS + OMPS. OSIRIS and MIPAS (as well as SCIAMACHY, GOMOS, and SMR) were included in the HARMOZ / Ozone_CCI merged data set used in WMO (2014), which is replaced here by the new SAGE + Ozone_CCI + OMPS data set. The most important differences between this study and Harris et al. (2015) are the different trend periods, the use of the new and improved Ozone_CCI, SAGE + OSIRIS (now drift-corrected) data sets, and the omission of the anomalous SAGE + GOMOS data set. The latter two data sets provided quite different trend estimates from each other, and from other data sets (compare Fig. 6 of Harris et al., 2015).

## 2.2 Time Series

Figure 1 shows annual mean ozone anomalies from the different satellite and ground-based data sets, averaged over three latitude bands, and for a level near 2 hPa or 42 km. Anomalies are relative to the 1998 to 2008 climatology of each individual data set. 1998 to 2008 was chosen as the reference period because ozone values were fairly constant over this period, and many instruments provide data for a substantial fraction of the period: SAGE II and HALOE until late 2005, ENVISAT instruments since late 2002, and Aura-MLS since August 2004. All data sets show clear ozone declines until the late 1990s and generally increasing ozone over the 2000 to 2016 period, especially at mid-latitudes. This observed evolution generally confirms expectations from model simulations by Chemistry Climate Models within their Validation-2 initiative (CCMVal-2, Eyring et al., 2010). Grey lines and shading in Fig. 1 give the CCMVal-2 multi-model mean and $\pm 2$ standard deviation envelope, obtained over all models (except outliers) and a 25 month sliding window. The simulations attribute the ozone decline until the late 1990s to increasing ODS loading, and predict positive ozone trends due to declining ODS loading since around 2000. In the simulations, the ozone increase is enhanced by overall cooling of the stratosphere due to increasing greenhouse gases over the entire 1978 to 2016 period (see also Jonsson et al., 2004; Randel et al., 2016).

All observational data sets show similar fluctuations from year to year, usually within 1 or 2% of each other. They also indicate similar long-term tendencies, usually within $\pm 2\%$ per decade of each other, and comparable to the CCMVal-2 simulations. Generally, the station data show larger variations than the zonal means from the satellite data. This is not surprising, given the sparser temporal sampling of most ground-based data (lidar, Umkehr, and FTIR all require clear sky). Also, the low density of stations will generally result in more variability compared to the smoother wide-band zonal means from the satellite records.

## 3 Multiple Linear Regression

Multiple linear regression (MLR) has become a standard method for deriving ozone trends (Bojkov et al., 1990; Reinsel et al., 2002; Newchurch et al., 2003; Chehade et al., 2014). MLR can be applied to monthly mean ozone anomaly time series $dO_3(i)$ of many months $i$. The anomalies are obtained by referencing the monthly mean $O_3(i)$ to the climatological mean for each calendar month $O_{3,Clim}(i \bmod 12)$.

$$dO_3(i) = \frac{O_3(i) - O_{3,Clim}(i \bmod 12)}{O_{3,Clim}(i \bmod 12)} \tag{1}$$

MLR attempts to reconstruct the observed anomalies as a linear combination of prescribed predictors $P_j(i)$, which account for known ozone variations, and residual noise $\epsilon(i)$.

$$dO_3(i) = c_0 + \sum_{j=1}^{n} c_j * P_j(i) + \epsilon(i) \tag{2}$$

Here our set of predictors $P_j$ includes a linear trend, a change of the trend in January 1997 (hockey stick, reflecting the increase of ozone depleting substances until the late 1990s, and their decline since), six proxies for the Quasi-Biennial Os-

cillation (QBO, equatorial zonal winds at 30 and 10 hPa, plus their modulations by sine and cosine with 12 month (=annual) period), and a proxy for the 11-year solar cycle, as in Reinsel et al. (2002). Like WMO (2014) or Harris et al. (2015), the present study also includes a proxy for stratospheric aerosol loading and for El-Nino / La Nina, which is most relevant for the tropical lower stratosphere (e.g., Oman et al., 2013). Table 4 summarizes the proxies used, and their sources. Other studies
may include further proxies for weather patterns and meridional ozone transports, such as circulation indices or eddy heat flux (Steinbrecht et al., 2001; Reinsel et al., 2005), but this was not done here. The coefficients $c_j$ are obtained by least squares fitting of the residuals, i.e. minimization of $\sum_i \epsilon^2(i)$. Typically, the residuals $\epsilon(i)$ are of the order of 1 to 10%, large enough to cover fit errors and measurement errors for each monthly mean.

If realistic uncertainties $\Delta O_3(i)$ are available for each monthly mean, the anomalies can be weighted by their inverse squared
uncertainty (high weight for low uncertainty), and the uncertainties $\Delta O_3(i)$ can be used to estimate the uncertainty $\Delta c_j$ of the fitted $c_j$. However, in many cases reliable uncertainties are not available for monthly means, because it is difficult to account correctly for all error terms, and for autocorrelation and covariance of the individual measurement errors (e.g., Toohey and von Clarmann, 2013; Damadeo et al., 2014). A time-invariant bias, for example, might be included in the monthly mean uncertainty, but it would be irrelevant for the long-term trend. So in many studies, including Reinsel et al. (2005), WMO (2014), Harris
et al. (2015), and this study, the pragmatic approach is to use the standard deviation of the fit residuals $\epsilon(i)$ for estimating the uncertainties $\Delta c_j$ of the fitted coefficients.

Strictly, the uncertainties from the MLR assume that the predictors are orthogonal, and that the residuals $\epsilon(i)$ are uncorrelated white noise. In practice, the predictors above are orthogonal enough (cross-correlations less than 0.3 for the long periods considered), and first order auto-correlation in the residuals is small ($|AC| \ll 0.3$). Still, to correct for first order auto-correlation
$AC$ (Reinsel et al., 2002), the $\Delta c_j$ are multiplied here by $\sqrt{(1+AC)/(1-AC)}$. Neglecting higher orders of autocorrelation might result in slightly underestimated uncertainties (Vyushin et al., 2007).

One problem with the "hockey stick" fit is that the slope of the declining trend and the time of the turning point have an influence on the slope of the second part of the "hockey stick" (Reinsel et al., 2002). To reduce this problem, a second step was introduced in WMO (2014), and this approach is also used here. First, Equation 2 is fitted for the entire long time series, e.g.,
from 1978 to 2016. The QBO, solar cycle, aerosol and El Nino effects resulting from this first fit are then subtracted from the ozone anomalies $dO_3(i)$. This provides time series of ozone residuals $O_{3,res}(i)$, which have most of the variability associated with QBO, solar cycle, aerosol and El Nino removed, but which still contain the long-term trend component, substantial remaining variability and the $\epsilon(i)$. The use of the entire 30 to 40 year long time-series in the first step is particularly important for a good estimate of the 11-year solar cycle effect, which cannot be estimated well with shorter records. Then, in a second
step, a simple linear trend is fitted to the $O_{3,res}(i)$. This trend can be fitted over any desired period, in this case the period 2000 to 2016. The second fit is not constrained by a "hockey stick" assumption, and has full freedom to react to the remaining ozone variations over the desired period.

## 4 Ozone Profile Trends

### 4.1 Trends for individual data sets

Figures 2 and 3 present the latitude-pressure cross-sections of 2000 to 2016 ozone trends, $TR$, (and uncertainties $\sigma$) obtained using the two-step method from the previous section for the satellite-based data sets from Section 2. In addition, the top right panel shows corresponding trends for the multi-model mean of the CCMVal-2 simulations. For the simulations, trend uncertainty was derived from the standard deviation of individual monthly anomalies from the multi-model mean (shaded envelope in Fig. 1). For the observations, trend uncertainty was derived from the fit residuals, as mentioned in Section 2. Although the two approaches differ, fit residuals for the observations and standard deviation of the simulated monthly anomalies have similar magnitude (compare Fig. 1), and the resulting trend uncertainties come out similar. The magnitude of all trends is represented by the color scale. Grey shading indicates regions where trends are not statistically significant (95% confidence level, $|TR| \leq 2\sigma$). All satellite data sets show significant ozone increases in the extra-tropical upper stratosphere, above 10 to 5 hPa (30 to 35 km). Some show significant ozone increases also in the tropical upper stratosphere. At levels between 50 and 10 hPa (20 and 30 km), trends are generally not significant, except for islands of significant trends near 20 hPa or 50 hPa in some data sets, mostly in the southern hemisphere. Most data sets (but not SAGE + MIPAS + OMPS) also show significant ozone decline in the tropical lowermost stratosphere below 100 hPa (16 km). However, satellite measurements in this region can have large uncertainties and need very careful consideration, both in tropics and extra-tropics (see also Tegtmeier et al., 2013).

The simulations (in the top right panel of Fig. 2) confirm that significant trends should be expected only in the upper stratosphere, between 10 and 0.5 hPa (30 to 55 km), especially in the extra-tropics. Exactly there, the observed data sets give significant increasing trends. Both magnitude - between 1 and 5% per decade - and latitudinal pattern - smaller increases in the tropics, larger increases at higher latitudes - are consistent between the satellite data sets and the simulations.

Figs. 2 and 3, therefore, provide substantial observational evidence for significant ozone increases in the upper stratosphere, consistent with model simulations based on declining ODS and decreasing temperatures in the upper stratosphere. Comparison of Figs. 2 and 3 with Figure 2-10 of WMO (2014) shows that the addition of three more years of data, as well as improved and additional data sets, have not changed the overall picture very much. Comparable patterns, but slightly smaller increases are also reported in Fig. 5 of Harris et al. (2015) for the 1998 to 2012 period (but only between 60°S and 60°N). Compared to that Figure, the SAGE + OSIRIS trends here have changed considerably because of improved data (see also Bourassa et al., 2017). The SWOOSH trends have increased in magnitude. The SAGE + GOMOS data, which had shown large, and probably unrealistic ozone decline polewards of 40° latitude in Fig. 5 of Harris et al. (2015) are not used here.

A specific look at zonal mean trends from all satellite and ground-based data sets is given in Fig. 4. The basis for these trend calculations are zonal band anomaly time series as in Fig. 1. In Fig. 4, almost all individual data sets show increasing ozone between 5 and 1 hPa, with trends between 0 and 4% per decade. For the 5 and 2 hPa levels, the plotted $\pm 2\sigma$ uncertainty bars (from the MLR) indicate that most individual trends are statistically significant (95% confidence level). Between 50 and 10 hPa (22 and 30 km), most data sets indicate small and non-significant trends. In the lowermost stratosphere, between 100 and

50 hPa (16 and 22 km), several data sets report ozone decreases, but these are generally not statistically significant. Differences between data sets are larger as well. Overall, Fig. 4 confirms significant ozone increases in the upper stratosphere from nearly all satellite and ground-based data sets, whereas ozone trends at lower levels are generally smaller and not significant.

## 4.2 From individual data set trends to the average trend

It is useful to obtain an average ozone trend profile from all individual trends. In WMO (2014) this was done by a weighted mean of all individual ground-based and satellite trends $TR(i)$. Each individual trend was weighted with its inverse squared uncertainty $(1/\sigma(i))^2$, so more uncertain trends have less weight. Individual uncertainties $\sigma(i)$ came from the regression (as in Section 3), and also included a 1 or 2% per decade drift uncertainty ($2\sigma$, depending on the instrument) added in quadrature. This standard weighted mean approach (SWM) was also one of the approaches used in Harris et al. (2015), however, with much

larger drift uncertainties (4 or 6% per decade, $2\sigma$). This resulted in larger overall uncertainty and in non-significant trends in Harris et al. (2015) compared to WMO (2014).

    One problem with the standard weighted mean is that its uncertainty does not depend on the spread of the individual trends (because of Gaussian error propagation). Therefore, Harris et al. (2015) also considered the joint distribution approach (J). There, the uncertainty of the mean trend is essentially given by the standard deviation $\sigma$ between the individual trends (which

includes possible instrument drifts), divided by $\sqrt{n}$, where $n$ is the number of data sets. Strictly, $n$ should be the number of statistically independent data sets. However, since most merged data sets use the same SBUV, SAGE, MLS, or OMPS instruments, these data sets are not independent. Also, trend calculation by multiple linear regression uses the same approach and the same proxies for all data sets, which may further reduce independence between the individual trend estimates.

    To be compatible with Harris et al. (2015), where standard weighted mean approach with large drift uncertainties and joint

distribution approach gave similar average trends (1 to 3% per decade in the upper stratosphere) and similar uncertainties (2 to 6% per decade, $2\sigma$), it was decided to also use the joint distribution approach for the average trend in the present study. Table 5 summarizes the methods used in the different studies to arrive at an average trend and its uncertainty.

    Fig. 5 shows the joint distribution average trends (black lines), obtained here by averaging the seven satellite data sets (GOZ-CARDS, SWOOSH, SAGE + OSIRIS, SAGE + CCI + OMPS, SAGE + MIPAS + OMPS, SBUV-NASA, and SBUV-NOAA).

All were given the same weight, but SBUV data were used only at levels above 40 hPa (23 km), because the lower SBUV layers mix stratospheric and tropospheric ozone information due to their very wide averaging kernels (see also Kramarova et al., 2013). Ground-based data were not included in the average trends, because of their sparser sampling (which would require small weights), and also to be compatible with Harris et al. (2015). Nevertheless, the ground-based trends, shown in Fig. 4, provide important independent verification of the satellite-based trends.

The (joint distribution) uncertainty bars in Fig. 5 (black error bars) give the full $\pm 2\sigma$ standard deviations between all 7 satellite-based trend estimates. Using these uncertainty bars in the Figure assumes only 1 independent realization ($n = 1$), and should give a very conservative uncertainty estimate for the mean trend, $TR$. Even with this conservative uncertainty estimate, significant increasing trends ($|TR| \geq 2\sigma$) appear in Fig. 5 for the 2 hPa level in the tropics and at northern mid-latitudes. Table 6 gives the same trend results, but now bold letters indicate trends, $TR$, that are significant with 95% confidence ($|TR| \geq 2\sigma/\sqrt{n}$),

assuming $n = 3$ independent realizations, or $n = 2$ below 40 hPa. In this less conservative case, significant increasing trends appear nearly everywhere above 10 hPa (30 km). As mentioned above, trends at 70 hPa (and below) differ more between data sets and should be considered with care. See also the large error bars below 50 hPa for the tropical latitudes in Fig. 5.

### 4.3 Comparison to previous studies

For comparison, the yellow lines and shading in Fig. 5 show average 1998 to 2012 trends and uncertainties from Harris et al. (2015, joint distribution case), and the blue lines give average 2000 to 2013 trends and error bars from WMO (2014). See Tables 3 and 5 for a summary of the different data sets and approaches. Overall, the updated 2000 to 2016 trend profiles (black lines) agree quite well with Harris et al. (2015) and with WMO (2014), especially when the overlapping error bars are considered. One difference is that the previous negative trend around 5-8 hPa in the tropics is not observed any more. The major
difference, however, is the substantially larger uncertainty range reported in Harris et al. (2015) for the upper stratosphere. As mentioned before, it is probably caused by two outlying data sets in Harris et al. (2015): 1.) An older version of the SAGE + OSIRIS data set, where the OSIRIS (V5.07) data suffered from a large drift (Hubert et al., 2016; Bourassa et al., 2017). 2.) A now outdated SAGE + GOMOS data set, which exhibited unrealistically low / negative trends at latitudes poleward of 45° (see Fig. 5 of  Harris et al., 2015). For levels above 5 hPa (35 km) and levels below 30 hPa (25 km), the new and improved
merged satellite data sets, and the additional years, provide substantially smaller trend uncertainties than Harris et al. (2015). In particular, a look at Fig. 4 indicates that individual trends from the merged satellite data sets and the ground-based instruments used here differ, in most cases, by less than 2 or 3% per decade, at levels above 50 hPa. This much better agreement indicates that previously large instrumental drifts and drift uncertainties (around 6% per decade for some data sets Harris et al., 2015; Hubert et al., 2016) have been reduced substantially since.
Compared to WMO (2014), the current work reports slightly larger uncertainty bars. This is expected, because the standard weighted mean uncertainty used in WMO (2014) did not consider the spread of the individual trends (as mentioned above), and also assumed statistical independence for all the data sets in the average.

The updated trend profiles in Fig. 5 also show excellent agreement with the CCMVal-2 simulations, with virtually no difference at levels above 50 hPa (20 km). The fact that all individual data sets in Fig. 4 indicate significant increases in the
upper stratosphere, the reduced uncertainty since Harris et al. (2015), the excellent agreement with the CCMVal-2 simulations in Fig. 5, and the good agreement with trend results from WMO (2014), all give enhanced confidence that ozone is indeed increasing in the upper stratosphere, and that at least part of that increase is due to declining ODS.

### 5   Conclusions

New and improved satellite data sets, and the addition of several years of data until the end of 2016, improve our confidence
that ozone in the upper stratosphere, between 5 and 1 hPa (35 to 48 km), has been increasing since 2000. Between 50 and 10 hPa (20 to 30 km) trends are small, and there are no clear indications for increasing (or decreasing) ozone. In the lowermost stratosphere, between 100 and 50 hPa (16 and 20 km), there might be an indication for decreasing ozone in the tropics and

at northern mid-latitudes. However, differences between data sets in this region are larger. Instrumental difficulties and large natural variability require more careful analysis of these possible ozone decreases.

Overall, the updated ozone profile trends are consistent with previous studies, e.g. with WMO (2014) and Harris et al. (2015), but average trend uncertainty in the upper stratosphere is reduced by a factor of two compared to Harris et al. (2015). Ozone increases at the 2 hPa (42 km) level are statistically significant with more than 90 or 95% confidence over a wide range of latitudes. In addition, the majority of all individual satellite and ground-based data sets also indicates significant ozone increases at levels above 10 hPa.

There are, however, remaining questions, for example regarding the merging of different instrumental records, the quality of the records in the lowermost stratosphere, or on the best methods for trend estimation and their detailed uncertainties. These issues are being addressed in the "Long-term Ozone Trends and Uncertainties in the Stratosphere" (LOTUS) initiative, which runs under the Stratosphere-troposphere Processes And their Role in Climate project (SPARC) of the World Climate Research Programme (WCRP), see http://www.sparc-climate.org/activities/ozone-trends/. The goals of LOTUS are to further improve the data sets, to better understand all relevant uncertainties, and to achieve a more complete and more precise picture of trends in the stratospheric ozone profile. What is also missing is a thorough quantification and attribution of the contributions from decreasing ozone depleting substances, from stratospheric cooling (due to increasing $CO_2$), and from transport changes to the observed profile trend. While this has been done in modelling studies (e.g. Jonsson et al., 2004; WMO, 2014), quantification of these factors on the basis of observations has not been done yet.

The update presented here, however, already gives strong indications that ozone in the upper stratosphere has been increasing over the last 15 years, and has begun to recover from man-made ozone depleting substances. Simulations show that this process will take many more decades. In order to verify that ozone recovery continues as expected, reliable long-term observations from multiple independent platforms will remain crucial for many years to come.

*Author contributions.* The paper was written by W. Steinbrecht, who also did the trend analysis, and is responsible for the NDACC lidar measurements at Hohenpeissenberg. L. Froidevaux, R. Fuller, H.J. Wang, and J. Anderson contributed the GOZCARDS data set, R.P. Damadeo, J.M. Zawodny, A. Bourassa, C. Roth, and D. Degenstein the SAGE + OSIRIS + OMPS data set, S. Frith, R.S. Stolarski, R.D. McPeters, and P.K. Bhartia the SBUV-NASA data set and OMPS data, J. Wild and C. Long the SBUV-NOAA data set, S. Davis and K. Rosenlof the SWOOSH data set, V. Sofieva, K. Walker, N. Rahpoe A. Rozanov, M. Weber and others the SAGE + ESA CCI + OMPS data set, and A. Laeng, T. von Clarmann, G. Stiller and N. Kramarova the SAGE + MIPAS + OMPS data set. NDACC lidar measurements were provided by S. Godin-Beekmann, T. Leblanc, R. Querel, and D.P.J. Swart. NDACC microwave measurements were given by I. Boyd, K. Hocke, N. Kämpfer, E. Maillard, L. Moreira, and G. Nedoluha. C. Vigouroux, T. Blumenstock, M. Schneider, O. García, N. Jones, E. Mahieu, D. Smale, M. Kotkamp, and J. Robinson provided the NDACC FTIR data. I. Petropavlovskikh and E. Maillard Barras processed the Umkehr data set. N.R.P. Harris, B. Hassler, D. Hubert, and F. Tummon contributed important input and discussions on trends, data sets, and their uncertainties.

*Acknowledgements.* The authors gratefully acknowledge the extremely important contribution from staff at the stations, who run and fix the ground-based systems, and from the many people involved in the satellite measurements. Funding by national and supra-national agencies is

also gratefully acknowledged. Some of the data sets were calculated with resources provided by the North-German Supercomputing Alliance (HLRN). The merged SAGE + ESA CCI + OMPS data set has been created in the framework of the ESA Ozone_cci project. OHP NDACC lidar measurements are funded by CNRS and CNES. Work performed at the Jet Propulsion Laboratory, California Institute of Technology, was done under contract with the National Aeronautics and Space Administration. NOAA supports and funds a major part of the Dobson Umkehr measurements, in collaboration with funding and work done by Meteoswiss, Switzerland; NIWA, New Zealand; the Australian Bureau of Meteorology; CNRS, France; and the University of Fairbanks, Alaska. We acknowledge the CCMVal-2 group for providing their model simulations. Fiona Tummon was supported by Swiss National Science Foundation grant number 20FI21_138017. We also thank the two reviewers for their helpful comments.

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

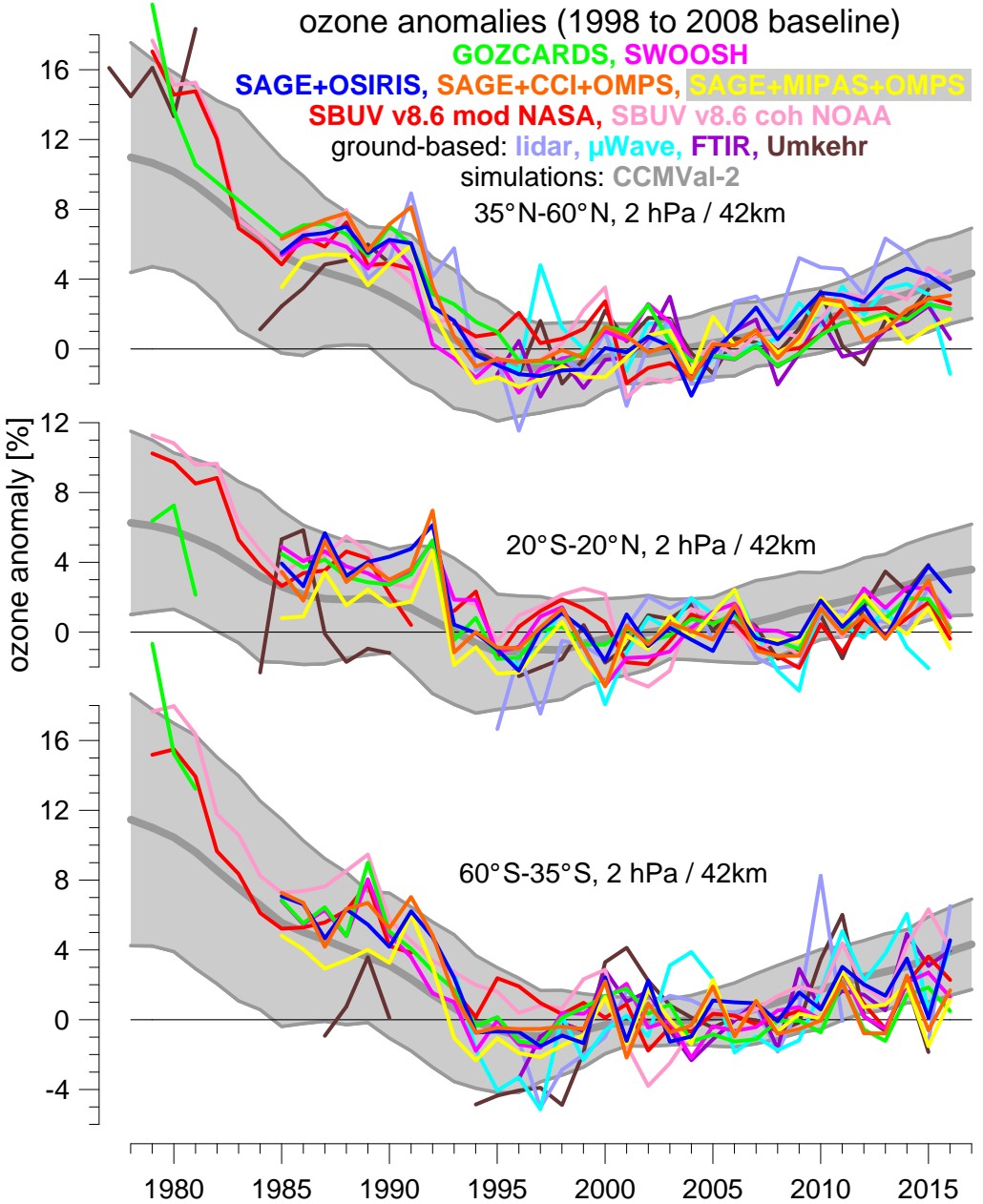

**Figure 1.** Annual mean ozone anomalies near 2 hPa or 42 km, as recorded by merged satellite data sets and ground-based stations. Anomalies are referenced to the 1998 to 2008 climatological annual cycle of each individual data set, and are averaged over the indicated zonal bands. Stations close to a zonal band are also included, i.e. NDACC lidar data from Table Mountain at $34.4°$N, NDACC FTIR data from Izaña at $28.3°$N and Wollongong at $34.4°$S, and Umkehr data from Perth at $34.7°$S are included in the respective mid-latitude bands. Due to contamination by volcanic aerosol after the eruptions of El Chichon and Mt. Pinatubo, Umkehr data are not used for the years 1982, 1983, and 1991 to 1993, and SBUV-NASA data are not available for 1992. Grey lines show the multi-model mean ozone anomalies from CCMVal-2 simulations (Eyring et al., 2010), with the grey shading giving the $\pm 2$ standard deviation envelope. Mean and standard deviation are taken over all models (except outliers) and within a 25 month sliding window.

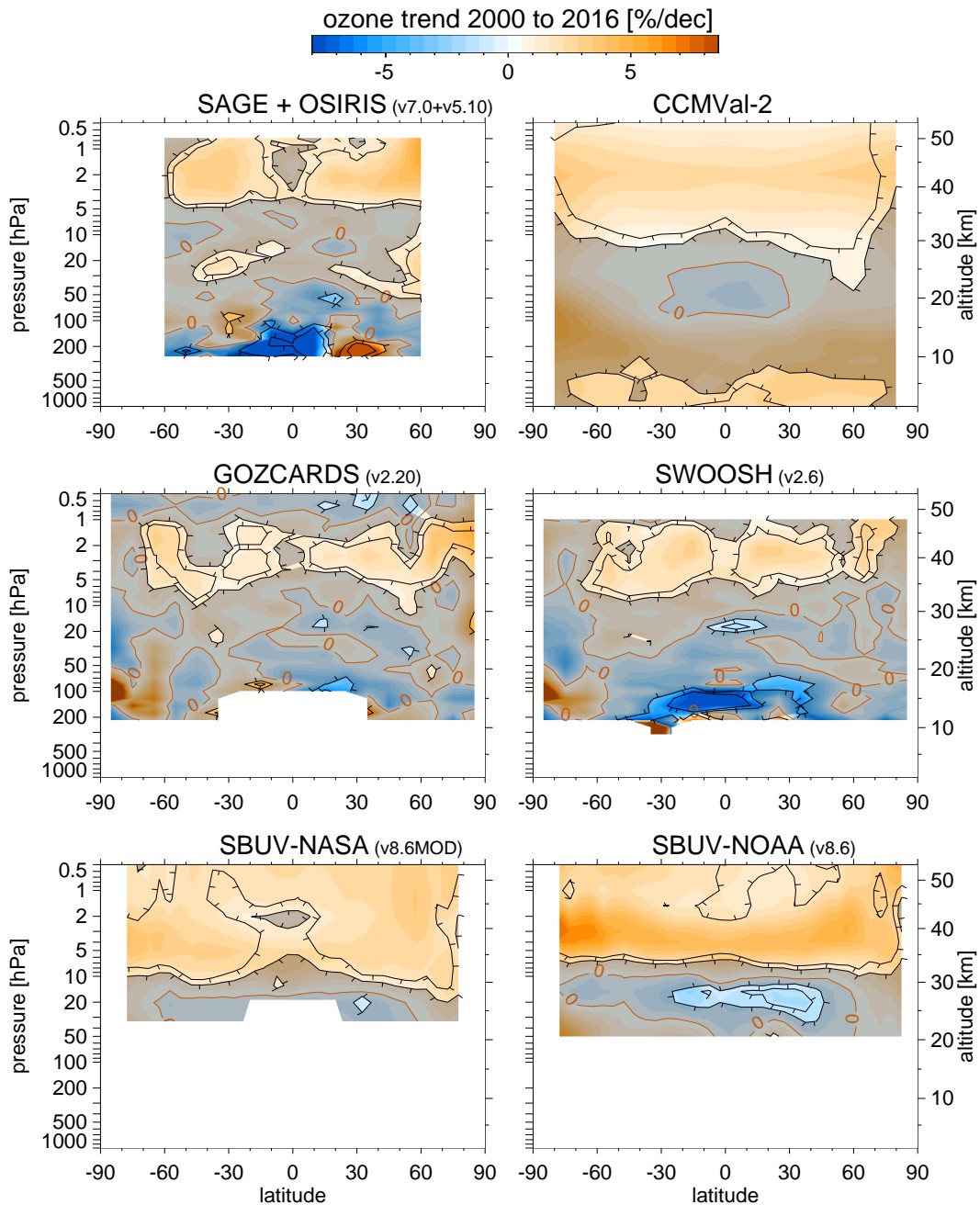

**Figure 2.** Latitude pressure cross section of 2000 to 2016 ozone trends $TR$ obtained by 2-step multiple linear regression (see text). The top right panel is for model simulations from the CCMVal-2 initiative. The other panels are for merged satellite data sets. The colour scale gives trend magnitude $TR$. Shading and isolines give the ratio of trend to trend uncertainty, $|TR|/\sigma$. Grey shading, in regions where $|TR| \leq 2\sigma$, indicates that trends are not significant at the 95% confidence level. The next isoline is at $|TR|/\sigma = 3$.

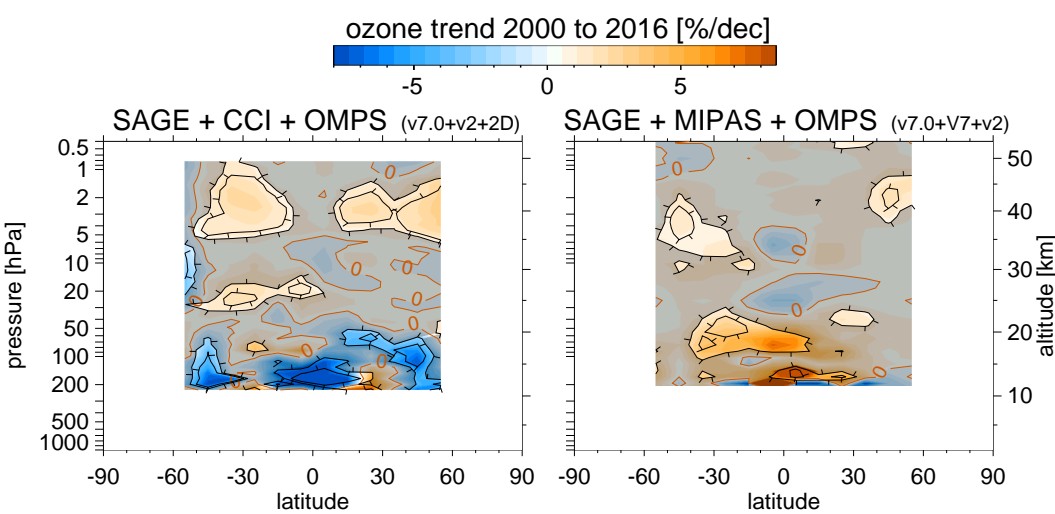

**Figure 3.** Same as Fig. 2, but showing the 2000 to 2016 ozone trends for the merged SAGE + ESA Ozone CCI + OMPS, and SAGE + MIPAS + OMPS data sets. The SAGE + ESA Ozone CCI + OMPS data set uses the OMPS 2D retrieval from U. Saskatoon. The SAGE + MIPAS + OMPS data set uses the OMPS v2 retrieval from NASA.

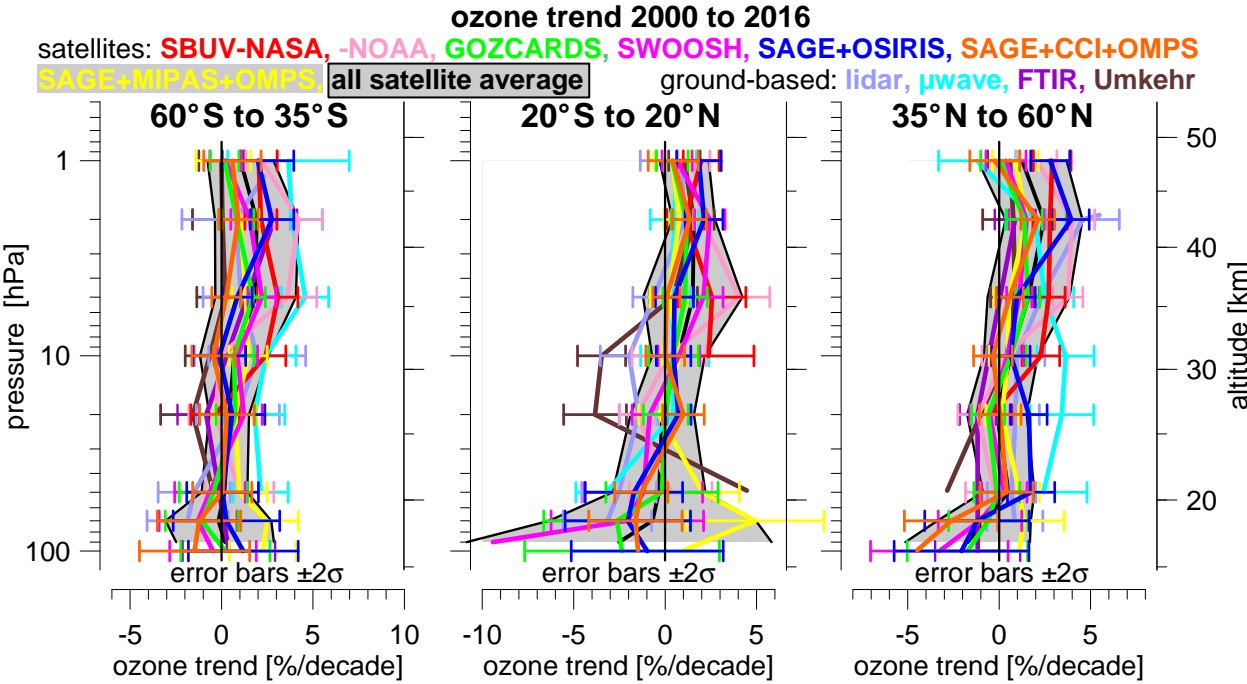

**Figure 4.** Vertical profiles of 2000 to 2016 ozone trends, obtained by 2-step multiple linear regression (see text), for different merged satellite and ground-based station data sets. Results are for the zonal bands 60°S to 35°S (left), 20°S to 20°N (center) and 35°N to 60°N (right). For the 60°S to 35°S zonal band, FTIR data from Wollongong (34.4°S), and Umkehr data from Perth (34.7°S) are included. For the 35°N to 60°N band, lidar data from Table Mountain (34.4°N) and FTIR data from Izaña (28.3°N) are included. SBUV and Umkehr data are not shown at/ below the 50 hPa level. Black lines and grey shading show the average trend and $\pm 2\sigma$ standard deviations of the 7 satellite based trends (see also Fig. 5 and text).

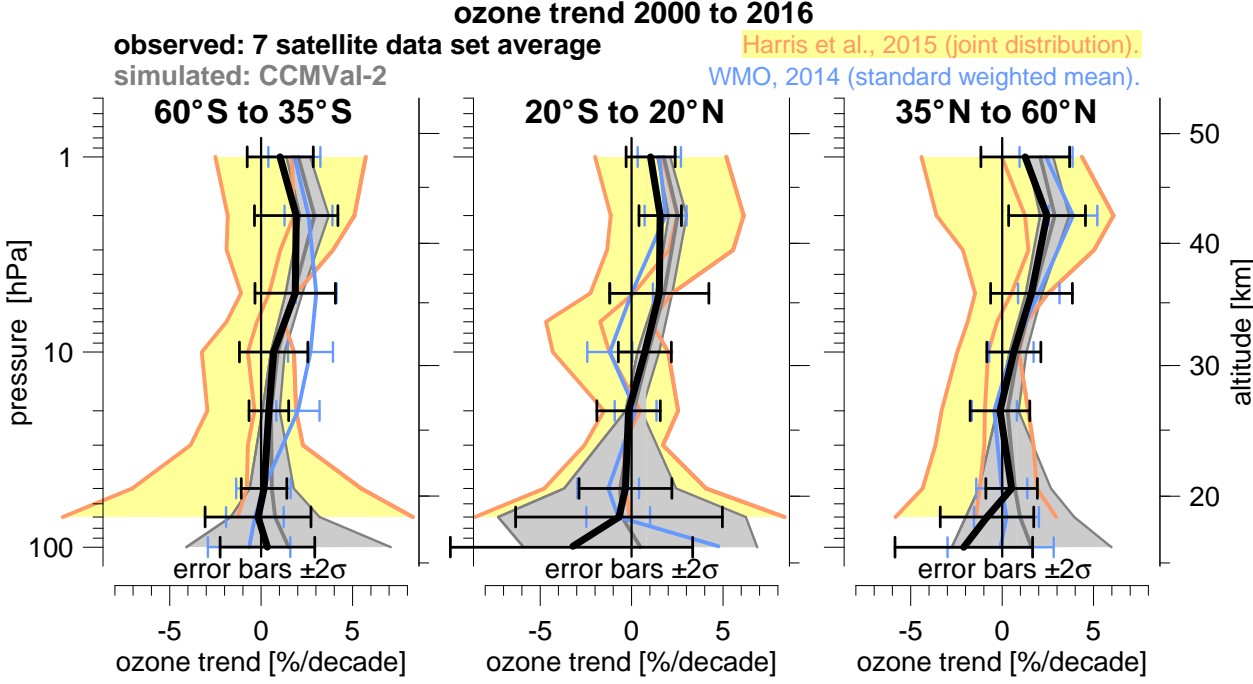

**Figure 5.** Same as Fig. 4, but giving the average 2000 to 2016 ozone trends (black lines) from seven merged satellite data sets (GOZCARDS, SWOOSH, SAGE + OSIRIS, SAGE + CCI + OMPS(2D), SAGE + MIPAS + OMPS(v2), SBUV-NASA, and SBUV-NOAA). SBUV trends are only used at levels above 40 hPa (23 km). For comparison, the 1998 to 2012 average ozone trend from Harris et al., 2015 (yellow lines and shading), and the 2000 to 2013 average ozone trend from WMO 2014 ( blue lines) are shown as well. In all cases, uncertainty bars or shading give $\pm 2\sigma$ uncertainty. For the 2000 to 2016 satellite average ozone trends (black error bars), and trends from Harris et al., 2015 (yellow shading) the uncertainty is derived from the standard deviation $\sigma$ between individual trends in the average (joint distribution case). For the WMO 2014 trends (blue error bars), the uncertainty of the standard weighted mean is given (see text for details). Grey lines and shading give the trend and $\pm 2\sigma$ uncertainty obtained from multi-model mean and standard deviation of the CCMVal-2 model simulations (see text in Section 4.1).

**Table 1.** Merged satellite data sets used in the present study. The URLs serve as an entry point only, and do not always provide the newest and most complete data set used here. See text for references.

| Name | Version(s) | from | to | URL |
|---|---|---|---|---|
| SBUV-NASA | v8.60MOD | 05/1970 [a] | 12/2016 | https://acd-ext.gsfc.nasa.gov/Data_services/merged/ |
| SBUV-NOAA | v8.60 | 11/1978 | 12/2016 | ftp://ftp.cpc.ncep.noaa.gov/SBUV_CDR/ |
| GOZCARDS | v2.20 | 02/1979 [b] | 12/2016 | https://gozcards.jpl.nasa.gov/ |
| SWOOSH | v2.6 | 10/1984 | 12/2016 | https://www.esrl.noaa.gov/csd/groups/csd8/swoosh/ |
| SAGE II + OSIRIS (+ OMPS)[c,d] | v7.0 + v5.10 (+ 2D[d]) | 10/1984 | 12/2016 | http://osirus.usask.ca/ |
| SAGE II + Ozone_CCI + OMPS[d] | v7.0 + v2 + 2D[d] | 10/1984 | 12/2016 | http://www.esa-ozone-cci.org/ |
| SAGE II + MIPAS + OMPS[e] | v7.0 + KIT v7 + v2[e] | 10/1984 [f] | 03/2017 | https://www.imk-asf.kit.edu/english/304_2857.php |

[a] gap from 05/1976 to 10/1978; [b] includes also SAGE I, but gap from 12/1981 to 09/1984, when SAGE II begins; [c] the SAGE + OSIRIS data set optionally includes OMPS data. These start in 04/2012, and give very similar trend results. However, to keep more independence between the various data sets, the version with OMPS data is not used here; [d] OMPS 2D retrieval from U. Saskatoon; [e] OMPS retrieval from NASA; [f] MIPAS high resolution data from 07/2002 to 03/2004, reduced resolution data from 01/2005 to 04/2012, gap in between.

**Table 2.** Stations and instruments used in the present study. Lidar, microwave and FTIR data are from the Network for the Detection of Atmospheric Composition Change (NDACC), and are originally available at http://www.ndacc.org. Umkehr data were provided by I. Petropavlovskikh.

| Name | latitude | longitude | instrument | from | to |
|------|----------|-----------|------------|------|-----|
| Fairbanks | 64.8°N | 147.9°W | Umkehr | 03/1994 | 09/2015 |
| Hohenpeissenberg | 47.8°N | 11.0°E | lidar | 09/1987 | 12/2016 |
| Bern | 46.9°N | 7.5°E | microwave | 11/1994 | 12/2016 |
| Payerne | 46.8°N | 7.0°E | microwave | 01/2000 | 12/2016 |
| Arosa | 46.8°N | 9.7°E | Umkehr | 01/1956 | 12/2015 |
| Jungfraujoch | 46.6°N | 8.0°E | FTIR | 05/1995 | 11/2016 |
| Haute Provence | 43.9°N | 5.7°E | lidar | 07/1985 | 10/2016 |
| Haute Provence | 43.9°N | 5.7°E | Umkehr | 01/1984 | 11/2015 |
| Boulder | 40.0°N | 105.3°W | Umkehr | 01/1984 | 12/2015 |
| Table Mountain | 34.4°N | 117.7°W | lidar | 02/1988 | 09/2016 |
| Izaña | 28.3°N | 16.5°W | FTIR | 03/1999 | 10/2016 |
| Mauna Loa | 19.5°N | 155.6°W | lidar | 07/1993 | 09/2016 |
| Mauna Loa | 19.5°N | 155.6°W | microwave | 07/1995 | 05/2015 |
| Mauna Loa | 19.5°N | 155.6°W | Umkehr | 01/1984 | 12/2015 |
| Wollongong | 34.4°S | 150.9°E | FTIR | 05/1996 | 11/2016 |
| Perth | 34.7°S | 138.6°E | Umkehr | 01/1987 | 12/2015 |
| Lauder | 45.0°S | 169.7°E | microwave | 10/1992 | 10/2016 |
| Lauder | 45.0°S | 169.7°E | lidar | 11/1994 | 12/2016 |
| Lauder | 45.0°S | 169.7°E | FTIR | 10/2001 | 12/2016 |
| Lauder | 45.0°S | 169.7°E | Umkehr | 02/1987 | 12/2015 |

**Table 3.** Comparison between principal data sets, trend periods, and regression method used in the present study, in Harris et al. (2015), and in WMO (2014). Here only major changes in data sets are indicated. Boldface indicates the most relevant differences.

| | this study | Harris et al. (2015) | WMO (2014) |
|---|---|---|---|
| SBUV-NASA | used | used | used |
| SBUV-NOAA | used | used | not used |
| GOZCARDS | used | old version | used |
| SWOOSH | used | old version | not used |
| Ozone_CCI | new version | not used | old version[a] |
| SAGE + OSIRIS | **new version[b]** | **old version[c]** | **not used** |
| SAGE + GOMOS | **not used** | **old version[c]** | **not used** |
| SAGE + MIPAS + OMPS | used | not available | not available |
| ground-based | used[d] | used[d] | used[e] |
| trend period | **2000 to 2016** | **1998 to 2012** | **2000 to 2013** |
| regression method | two steps | hockey-stick | two steps |

[a] called HARMOZ. [b] drift-corrected OSIRIS data. [c] OSIRIS and GOMOS data had significant drifts (Tegtmeier et al., 2013; Harris et al., 2015; Hubert et al., 2016). [d] ground-based trends as independent verification, but not included in average trend. [e] ground-based trends (weighted) included in average trend.

**Table 4.** Proxy time series used for the multiple linear regression in Eq. 2 in this study.

| Proxy | description | URL |
| --- | --- | --- |
| trend | linear increase over entire time period. | |
| change of trend | "hockey stick": 0 before 01/1997, linear increase after. | |
| QBO | 10 and 30 hPa equatorial zonal wind from Singapore radio-sondes, as compiled by FU Berlin. To account for annual variation and phase of the QBO influence, QBO(10), QBO(10)$\cdot\sin(j)$, QBO(10)$\cdot\cos(j)$, QBO(30), QBO(30)$\cdot\sin(j)$, and QBO(30)$\cdot\cos(j)$ are fitted, with $j = 2\pi \cdot (\text{month} \bmod 12)/12$. | http://www.geo.fu-berlin.de/en/met/ag/strat/produkte/qbo/index.html |
| Solar Cycle | solar radio flux at 10.7 cm, observed at Penticton, Canada. | ftp://ftp.geolab.nrcan.gc.ca/data/solar_flux |
| El-Nino | Multivariate ENSO index from Wolter and Timlin (2011). | https://www.esrl.noaa.gov/psd/enso/mei/ |
| Aerosol | stratospheric aerosol optical depth following Sato et al. (1993). | https://data.giss.nasa.gov/modelforce/strataer/ |

**Table 5.** Approaches taken to obtain the average trend and its uncertainty estimate in the present study, in Harris et al. (2015), and in WMO (2014). Boldface indicates the approach used for the results plotted in Fig. 5.

| | this study | Harris et al. (2015) | WMO (2014) |
|---|---|---|---|
| standard weighted mean (SWM) | not used | used | **used** |
| assumed drift uncertainty ($2\sigma$) | not used | 4 or 6% decade | **1 or 2% decade** |
| number of data sets in SWM | – | 6[a] | **9**[b] |
| joint distribution (J) | **used** | **used** | not used |
| number of data sets in J | **7**[a] | **6**[a] | – |

[a] only satellite based data sets included in average. [b] ground-based data sets were also included. Ozone sondes not used at levels higher than 10 hPa / 31-km.

**Table 6.** Average 2000 to 2016 ozone profile trends, obtained from individual trends for the GOZCARDS, SWOOSH, SAGE + OSIRIS, SAGE + CCI + OMPS(2D), SAGE + MIPAS + OMPS(v2), SBUV-NASA, and SBUV-NOAA satellite data sets. Given are mean trend $TR$ and standard deviation $1\sigma$ of the individual trends, in percent per decade. Bold numbers indicate average trends $TR$ larger than $2\sigma/\sqrt{3} \approx 1.15\sigma$, i.e. statistically significant with 95% confidence, assuming that the 7 data sets give 3 independent realizations of individual trends $TR(i)$. The SBUV-NASA and SBUV-NOAA data sets are used only at levels above 40 hPa (23 km). Therefore, $2\sigma/\sqrt{2} \approx 1.41\sigma$ is applied as threshold for bold face at the 50 and 70 hPa levels.

| level | 60°S to 35°S | | 20°S to 20°N | | 35°N to 60°N | | 60°S to 60°N | |
|---|---|---|---|---|---|---|---|---|
| (hPa) | $TR$ | $1\sigma$ | $TR$ | $1\sigma$ | $TR$ | $1\sigma$ | $TR$ | $1\sigma$ |
| 1 | **1.0** | 0.9 | **1.0** | 0.7 | **1.3** | 1.2 | **1.1** | 0.7 |
| 2 | **1.9** | 1.1 | **1.6** | 0.6 | **2.5** | 1.1 | **1.8** | 0.6 |
| 5 | **1.9** | 1.1 | 1.5 | 1.4 | **1.6** | 1.1 | **1.6** | 1.2 |
| 10 | 0.7 | 0.9 | 0.7 | 0.7 | 0.6 | 0.7 | 0.8 | 0.7 |
| 20 | 0.4 | 0.5 | -0.2 | 0.9 | -0.1 | 0.8 | 0.0 | 0.7 |
| 50 | 0.2 | 0.6 | -0.3 | 1.3 | 0.5 | 0.7 | 0.0 | 0.8 |
| 70 | -0.2 | 1.4 | -0.7 | 2.8 | -0.8 | 1.3 | -0.6 | 1.9 |

All values are % per decade.