# Peer review of "An update on ozone profile trends for the period 2000 to 2016"

_Atmospheric Chemistry and Physics, 2017_

## Referee Comment (RC1) · J. Staehelin (Referee) · 31 May 2017

The paper provides an update and a valuable summary of recent studies of upper stratospheric ozone trends. This work is relevant in the context of the Montreal Protocol (1987) as positive ozone trends in the upper stratosphere (in extra tropics) are viewed as clear indication of decreasing anthropogenic ozone depletion caused by ozone depleting substances (ODS) which is also seen in numerical simulation. The paper is an extension of earlier work of last WMO/UNEP Ozone Assessment (2014). Quasi global ozone trends need to be determined from merged satellite series, but it turned out that the construction of merged satellite series from individual measurement series is a major challenge for the community both in terms of concept and available data. An additional challenge is the way how to combine different merged satellite series to obtain appropriate long-term trends series including determination of correct

uncertainties. Particularly useful is in my view the synopsis of recent activities including work of the past SPARC-activity SI2N (e.g. Harris et al., 2015). General comment: I recommend to discuss in some more detail the (conceptual) differences between Harris et al., 2015 and WMO, 2014 (2014), see below. Specific comments: Abstract: 1. p. 2, line 10/11: I agree that a "detailed attribution of the observed increases to declining ozone depleting substances and to stratospheric cooling" is required: From a formal point of view a suggest to mention this point in the conclusions as well. Introduction: 2. p. 2, Line 21/22: I don't believe, that the Montreal protocol was (only) signed because of the ozone hole, this is a too strong oversimplification for me: Indeed the ozone hole was very important to enhance public awareness but in the same period the results of the International Ozone Trend Panel Report were elaborated showing first time significant negative trends in northern mid latitudes which was certainly important for the signature (and gradual strengthening) of the Montreal Protocol too. Ozone profile data records: 3. I suggest to clarify whether additional data were used compared to WMO (2014) and for which data sets important revisions were made. 4. I think it would be useful to clarify whether (all) satellite merged series used in this paper were used in Harris et al. (2015) and vice versa: I think in this paper SAGE-GOMOS merged series are not used. Is there a particular reason not to use these data ? please explain 5. Second last para on page 4: I would have preferred to put the sentence "Table 2 summarizes the ground-based stations used in the present study" at the beginning of the para. 6. p. 4, line 29: I suggest to extend the paragraph staring on p. 4, line 29 about the comparison of Hubert et al., 2016: Which NDACC measurements were used ? 7. Figure 1, legend: I am not sure, whether Umkehr measurements belong to "NDACC ground-based stations" – Umkehr measurements at least started earlier than NDACC exists. 8. Figure 1: Is there an explanation why upper stratospheric ozone decrease in extra tropics in the first years of the 1980s seems considerably larger in available measurements than in numerical simulations ? 9. Figure 1: what's the reason for missing data in the black curve 1983-1985 in the upper panel (northern midatitude) ? are the Umkehr data missing ? 10. p. 5, line 7 ff: The reference Eyring et al., 2010

seems rather old. Are no more recent publications available ? The data after 2010 are predictions in Eyring et al., 2010 Ozone profile trends: 11. Fig. 2: How is the significance levels determined for numerical simulations ? Is this (directly) comparable with significant trends in measurements ? Please explain From individual data sets trends to the average trend: 12. p. 7., line 22 ff: The use of weighting with inverse squared uncertainty might be viewed as scientifically arbitrary. I believe, that weighting with inverse squared uncertainty of the individual data series tends to increase magnitude of trends of the ensemble. Please comment 13. I suggest to extend the first para of this section, I think this discussion is important for the community 14. Fig. 5: I am wondering whether it is justified to show the uncertainty of Harris et al., 2015 (yellow shading). The main result of this study seems to me that the uncertainty of Harris et al, 2015 no longer corresponds to the present knowledge which is shown in Table 4 namely because of longer series and improved data.

―――――――――――――――――――――

---

## Referee Comment (RC2) · Anonymous Referee #2 · 4 Jul 2017

Overall, the manuscript represents an important and timely update to the question of the magnitude and significance of trends in stratospheric ozone, which are an important indicator of the efficacy of the Montreal Protocol. The paper is generally well-written and the analysis is robust. There needs to be, however, a clearer, more focused discussion of the relationship between this paper and Harris et al. [2015] as a thread that one can follow throughout the paper. Instead, I found myself having to re-read Harris et al. to understand the primary difference between their analysis and WMO2014, and then having to read between the lines here to understand whether this was an apples-to-apples type update to the Harris et al. analysis or a simple extension of the WMO2014 analysis. The issue is easily fixed by being more clear about the differences between WMO2014 and Harris et al. (2015) early in the paper and by guiding the reader to

understand that the analysis here uses the same uncertainty assessment (based on the J-distribution) that led to the larger uncertainties in Harris et al. (2015) relative to WMO2014. I therefore recommend publication in Atmospheric Physics and Chemistry with minor revisions.

Specific Comments:

Line 2: It would be clearer to say "ground-based data [collected or measured] by four techniques....

Lines 6-8: This sentence should be rearranged to clarify that "more years of observations and updated data sets" refers to a comparison to WMO 2014 and Harris et al. 2015. (Suggested: "This study confirms positive trends already reported in. . .using three to four more years. . .."

Lines 8-10: Here it would be helpful if the authors were specific regarding the reduction in uncertainty relative to Harris et al. that is the result of the improved datasets, which have a lower inferred drift.

Line 29: It seems to me that a brief mention of confounding factors is warranted here. We do indeed expect ozone to increase, but it is important to be clear that variability and trends in the circulation, temperature changes, etc, can easily mask those increases and lead to large uncertainties on calculated trends even for ideal data records.

Lines 4-6: A more in-depth discussion of the differences between Harris et al. and Hubert et al. and the WMO 2014 results is needed here. What drove the larger uncertainties in those studies relative to WMO? Are the differences in uncertainties something that can be at least partially addressed by longer and / or improved records?

Section 2: The last paragraph of the introduction mentions "improved and additional

datasets". A brief summary of which datasets have been improved or added to the analysis since WMO 2014 would be beneficial in Section 2.

Line 13: For people with less familiarity with these stratospheric trend studies, it might not be clear why there is a focus on records starting before 1990 given that your analysis looks at the 2000-2016 period. Lines 23-26: This is confusing if one is unaware that there were two SAGE instruments. Line 29: Improved in what way?

Lines 1-3: A brief mention of the different assumptions used in SWOOSH and GOZ-CARDS would be helpful to a reader trying to understand how independent these datasets are.

Lines 17-28: The role of the ground-based data in this study is not quite clear to me. As far as I understand, they are not used in the average trend analysis, but they are also not used to quantitatively evaluate individual satellite records. It would be helpful if the authors could provide the rationale for their inclusion.

Lines 25-27: Is the fact that they are coherent over a wide range of altitudes and latitudes based on high vertical resolution profiles from satellites? Or on models? A reference would be useful here, since Harris et al. (2015) did not use ground-based profiles because of concerns about representativeness and the "coherency over latitude" would seem to negate that issue.

Lines 29-35 and Page 5, lines 1-3: These drifts, while not statistically significant, are of the same order of magnitude, if not larger than, the trends reported here. It is clear from the abstract that a more detailed analysis of these drifts is being undertaken by LOTUS, but a somewhat deeper discussion of their relevance to results presented here is needed, particularly given that the differences between Harris et al. and these results seems to stem in part from lower drift in the records used here.

Lines 5-6: What is the rationale for using the 1998-2008 climatology for normalization?

Lines 14-16: It seems this point could be made more clearly by referring to sparser spatial and temporal sampling rather than "sparser sampling" meaning temporal and "geophysical differences" referring to the spatial sampling.

Figure 1: Presumably the grey line in Figure 1 refers the multi-model mean of the CCMVal2 models? This point should be clear in the caption and in the text in Section 2, and the authors should consider providing the full envelope of the models, as the range is fairly large.

Line 27: Is there something missing here between "solar cycle" and "Reisel"?

Section 3: It is clear from Figure 1 that there are data gaps in at least some of the ground-based records. How are these handled in the trend analysis?

Lines 18-25: I found this explanation confusing. It is unclear to me from this description how the first 2 regression terms are used. This seems to imply that only the last 4 terms are used and then a linear trend for 2000-2016 is fitted to the remaining residuals – if so then why are the first 2 regression terms included at all?

Lines 20-23: On what years of data was the initial regression step performed?

Lines 2-3: The authors might want to refer to the Tegtmeier et al. paper on the SPARC Data Initiative ozone climatologies here.

Figure 3: What uncertainty was used for the CCMVal 2 results? Is it based on the model range for all of the models or just on the ensemble mean? Please specify.

Line 3: Why was SBUV only used above 40 hPa?

[Figure]

Figure 5: The caption needs several clarifications. It states that "uncertainty bars and yellow shading" give the +/- 2 x sigma values for all individual trends and seems to refer to the datasets used here (though it is unclear how both the uncertainty bars and yellow shading show the uncertainties for a single dataset), but then states that the yellow lines and shading show results from Harris et al. The WMO trend is apparently shown, but the color is not specified – is it the blue line? Finally, the clarification is again required for the model simulations – is this the ensemble mean? How are the uncertainties derived? I think perhaps things could be clarified if the sentence about uncertainty bars and shading were moved later in the paragraph.

Lines 20-21: Strictly speaking, a drift analysis requires comparison to independent datasets. It is unclear whether the authors are saying here that such an analysis has been performed and that the drift in the upper stratosphere has been determined to be 1-2% rather than the 6% used in Harris et al., or whether they are simply relying on the J-distribution analysis to argue for a small drift estimate. It is also unclear how this estimate relates to the estimates provide in Section 2, bottom of page 4, which describe drift estimates of 2-5% for the individual satellite records that make up the merged datasets.

Lines 16-21: For completeness, a brief discussion of the attribution of trends should be provided here.

---

## Author Comment (AC1) · 11 Aug 2017

**acp-2017-391: Steinbrecht et al. "An update on ozone profile trends for the period 2000 to 2016"**

**Response to Reviewers**

*We thank both reviewers for their positive reception of the manuscript, and for the helpful and constructive comments. The revised version of the manuscript addresses all their comments. Our detailed response to each comment follows below.*

The reviewers comments are given in normal typeface, *our responses are italicized and in red.*

The two major points raised by both reviewers were
1.) To improve and expand the discussion of differences and similarities between the present study, WMO (2014), and Harris et al. (2015).
2.) To clarify how trend uncertainty was determined for the CCMVal-2 multi-model simulations.

*To address these major points we have*
*1.) expanded the discussion of similarities and differences between the approach here, in WMO 2014, and Harris et al. 2015 throughout the paper. Additional text and explanation have been added in many parts, e.g. in abstract, introduction and discussion. We have also added two tables pointing out differences between the underlying data sets. Sub-sectioning of the text has also been increased to more clearly point out these comparisons.*
*2.) changed Figure 1 and added the ±2 standard deviation range of modelled anomalies, which is used to estimate trend uncertainty of the model simulations. We have also added a few explaining sentences to the beginning of Section 4 "Ozone Profile Trends" and the captions of Figures 2 and 5.*

*In addition to the changes outlined below, which were mostly minor textual changes, there were also a few other small text changes and changes in references. The description of QBO proxies, which in fact include an annual cycle, was also corrected. The original description was not correct.*

**Responses to Reviewer 1**

General comment:
I recommend to discuss in some more detail the (conceptual) differences between Harris et al., 2015 and WMO, 2014 (2014), see below.

*This is also suggested by reviewer 2. Consequently, the discussion of similarities and differences between the approach in this paper and in WMO 2014 and Harris et al. 2015 has been expanded throughout. Additional text and explanation have been added in many parts of the manuscript, e.g. in abstract, introduction and discussion. We have also added two tables pointing out differences between the underlying data sets and the way to determine average trend and its uncertainty.*

Specific comments:

Abstract:

1. p. 2, line 10/11: I agree that a "detailed attribution of the observed increases to declining ozone depleting substances and to stratospheric cooling" is required: From a formal point of view a suggest to mention this point in the conclusions as well.

*Good point, we have added a few sentences to the conclusions. Also suggested by reviewer 2.*

Introduction:
2. p. 2, Line 21/22: I don't believe, that the Montreal protocol was (only) signed because of the ozone hole, this is a too strong oversimplification for me: Indeed the ozone hole was very important to enhance public awareness but in the same period the results of the International Ozone Trend Panel Report were elaborated showing first time significant negative trends in northern mid latitudes which was certainly important for the signature (and gradual strengthening) of the Montreal Protocol too.

*We have reworded the paragraph to avoid this oversimplified and incorrect impression.*

Ozone profile data records:
3. I suggest to clarify whether additional data were used compared to WMO (2014) and for which data sets important revisions were made.

*As also suggested by reviewer 2, the discussion of similarities and differences between the data / approach in this paper and WMO 2014 and Harris et al. 2015 has been expanded throughout. A table, additional text and explanation have been added.*

4. I think it would be useful to clarify whether (all) satellite merged series used in this paper were used in Harris et al. (2015) and vice versa: I think in this paper SAGE-GOMOS merged series are not used. Is there a particular reason not to use these data ? please explain

*Same as point 3 above.*

5. Second last para on page 4: I would have preferred to put the sentence "Table 2 summarizes the ground-based stations used in the present study" at the beginning of the para.

*OK. Done.*

6. p. 4, line 29: I suggest to extend the paragraph staring on p. 4, line 29 about the comparison of Hubert et al., 2016: Which NDACC measurements were used ?

*Good point. We have added the information that only ozone sondes and lidars were used.*

7. Figure 1, legend: I am not sure, whether Umkehr measurements belong to "NDACC ground-based stations" – Umkehr measurements at least started earlier than NDACC exists.

*OK, while Dobson and Brewer instruments are NDACC-associated for total ozone, their Umkehr profiles are not really NDACC. We have reworded the legend.*

8. Figure 1: Is there an explanation why upper stratospheric ozone decrease in extra tropics in the first years of the 1980s seems considerably larger in available measurements than in numerical simulations ?

*As far as we know, there is no accepted clear-cut explanation. A number of factors can play a role:*

*a.) The SAGE I data (before 1982) have very sparse sampling, and have an altitude shift that is not very well known, and might be not well corrected for.*
*b.) the Umkehr data are also very sparse and have poor sampling. So the only "remaining" record is Nimbus 7 SBUV, which may or may not have a drift, and which may or may not be matched well to the later SBUVs. Another factor is the solar-cycle which is not simulated by all models, and not necessarily modeled well in the remaining models.*

*With the model envelope now given in the revised Figure 1, the difference in the earlier years is not as striking / clear cut anymore.*
*Because the early data and the trends before 1997 are not really at the focus of this study, we decided to not add discussion of this feature. As indicated above, we feel such a discussion would be inconclusive and might be more confusing than helpful.*

9. Figure 1: what's the reason for missing data in the black curve 1983-1985 in the upper panel (northern midatitude) ? are the Umkehr data missing ?

*In the post El-Chichon and Pinatubo years (1982, 1983, 1991 to 1993) the Umkehr data were heavily contaminated by volcanic aerosol and SBUV-NASA data are not available for 1992. Therefore data from these years were not used. We have added a corresponding sentence to the caption of Fig. 1.*

10. p. 5, line 7 ff: The reference Eyring et al., 2010 seems rather old. Are no more recent publications available ? The data after 2010 are predictions in Eyring et al., 2010 Ozone profile trends:

*Eyring et al. 2010 is still the most comprehensive / relevant reference for the CCMVal2 simulations. The more recent CMIP5 / ACCMIP interactive ozone simulations are only becoming available now, and were not available for the manuscript. For the upper stratosphere region of interest here changes between CCMVal1 and CCMVal2 were minimal, and no significant changes are expected from CMIP5 / ACCMIP. No change in the reference.*
*Technically, the CCMVal2 data after 2010 are "predictions". However, we feel that it is not necessary to introduce this technical term here in the text, and feel that describing them as "simulations" is adequate. No change to text.*

11. Fig. 2: How is the significance levels determined for numerical simulations ? Is this (directly) comparable with significant trends in measurements ? Please explain

*Trend uncertainty for the simulations is determined using the standard deviation of the simulated anomalies (=uncertainty of each data point). For the observations it is determined from the fit residuals. While not directly comparable, both approaches give similar results, because standard deviation of simulated anomalies and fit residuals are of comparable magnitude. We have added a few sentences to explain this to the discussion in the first paragraph of Section 4 "Ozone Profile Trends". Responding to similar questions by reviewer 2, we have also changed Figure 1 and added the ±2 standard deviation range of modelled anomalies, which essentially gives the uncertainty of the model simulated trends.*

**From individual data sets trends to the average trend:**

12. p. 7., line 22 ff: The use of weighting with inverse squared uncertainty might be viewed as scientifically arbitrary. I believe, that weighting with inverse squared uncertainty of the individual data series tends to increase magnitude of trends of the ensemble. Please comment

*We do not agree with these statements of the reviewer, and we have not changed the text here. Our reasoning is as follows:*

*a.) Weighted mean was used in WMO 2014 and Harris et al. 2015. We cannot change that, and there is no need to criticize that here. In the current paper, a weighted mean is not used anymore, as is discussed in the same paragraph.*

*b.) Weighting with inverse squared uncertainty is, in fact, standard procedure and makes perfect sense in many situations:*

*Assume that $n1$ and $n2$ samples are taken from the same random distribution with mean $X$ and standard deviation $\sigma$. We can then expect the mean of samples $n1$ to be $X1$ with uncertainty of the mean $dX1 = \sigma/\sqrt{n1}$, and the mean of samples $n2$ to be $X2$ with uncertainty of that mean $dX2 = \sigma/\sqrt{n2}$. Using both samples together as one will give the mean $X3 = (n1*X1 + n2*X2)/(n1+n2)$ with uncertainty of that mean $dX3 = \sigma/\sqrt{(n1+n2)}$.*

*This is precisely what is achieved by weighting $X1$ and $X2$ with their inverse squared uncertainties $n1/\sigma^2$ and $n2/\sigma^2$ ($\sigma^2$ falls out in the equation for $X3$). Gaussian error propagation ($dX3^2 = (\partial X3/\partial X1)^2 dX1^2 + (\partial X3/\partial X2)^2 dX2^2$;  independent samples!) also gives $dX3^2 = (n1/(n1+n2))^2 \sigma^2/n1 + (n2/(n1+n2))^2 \sigma^2/n2 = \sigma^2/(n1+n2)$, or $dX3 = \sigma/\sqrt{(n1+n2)}$ – exactly what is needed. Therefore, weighting by inverse squared uncertainty makes good sense. In particular it makes sure that more uncertain data points / samples carry less weight.*

*Why would weighting by inverse squared uncertainty increase the magnitude of the average trend? That would only happen, if larger trends were associated with larger weights, in this case smaller uncertainty. There is no reason why this would be the case here.*

*We, therefore, disagree with these statements of the reviewer, and we have not changed our text.*

13. I suggest to extend the first para of this section, I think this discussion is important for the community

*We assume that the referee means the first paragraph of Section 4.2 "from individual trends to average trend". As suggested by reviewer #2, we have modified the manuscript in several places, to give a more coherent discussion of the trends obtained here versus the trends in WMO 2014 and Harris et al. 2015. These text changes and the new additional table, in our opinion, also address this comment.*

14. Fig. 5: I am wondering whether it is justified to show the uncertainty of Harris et al., 2015 (yellow shading). The main result of this study seems to me that the uncertainty of Harris et al, 2015 no longer corresponds to the present knowledge which is shown in Table 4

*While we agree that the large uncertainty reported in Harris et al. 2015 is probably outdated, we still feel that it makes sense to show it, and to compare it with the new results. One of the results of our study is that the uncertainty of Harris et al. 2015 are conservative and probably too large! Therefore, to us, it makes perfect sense to show these uncertainties in the Figure. No changes.*

**acp-2017-391: Steinbrecht et al. "An update on ozone profile trends for the period 2000 to 2016"**

**Responses to Reviewer 2**

General Comment:
… The paper is generally well-written and the analysis is robust. There needs to be, however, a clearer, more focused discussion of the relationship between this paper and Harris et al. [2015] as a thread that one can follow throughout the paper. Instead, I found myself having to re-read Harris et al. to understand the primary difference between their analysis and WMO2014, and then having to read between the lines here to understand whether this was an apples-to apples type update to the Harris et al. analysis or a simple extension of the WMO2014 analysis. The issue is easily fixed by being more clear about the differences between WMO2014 and Harris et al. (2015) early in the paper and by guiding the reader to understand that the analysis here uses the same uncertainty assessment (based on the J-distribution) that led to the larger uncertainties in Harris et al. (2015) relative to WMO2014. …

*We thank the reviewer for this overall very positive, helpful and detailed review. As already mentioned in the reply to reviewer 1, the discussion of similarities and differences between this paper, WMO 2014 has Harris et al. 2015 has been expanded throughout the revised manuscript. We have also added two tables to clarify this.*

Specific Comments:
Line 2: It would be clearer to say "ground-based data [collected or measured] by four techniques....

*OK, we have added "measured".*

Lines 6-8: This sentence should be rearranged to clarify that "more years of observations and updated data sets" refers to a comparison to WMO 2014 and Harris et al. 2015. (Suggested: "This study confirms positive trends already reported in. . .using three to four more years. . .."

*OK, this has been reworded.*

Lines 8-10: Here it would be helpful if the authors were specific regarding the reduction in uncertainty relative to Harris et al. that is the result of the improved datasets, which have a lower inferred drift.

*These are all good points, and we have reworded the abstract accordingly.*

Line 29: It seems to me that a brief mention of confounding factors is warranted here. We do indeed expect ozone to increase, but it is important to be clear that variability and trends in the circulation, temperature changes, etc, can easily mask those increases and lead to large uncertainties on calculated trends even for ideal data records.

*OK. Masking / confounding factors are now mentioned, with references.*

Lines 4-6: A more in-depth discussion of the differences between Harris et al. and Hubert et al. and the WMO 2014 results is needed here. What drove the larger uncertainties in those studies

relative to WMO? Are the differences in uncertainties something that can be at least partially addressed by longer and / or improved records?

*Good point. We have added discussion and have expanded the paragraph.*

Section 2: The last paragraph of the introduction mentions "improved and additional datasets". A brief summary of which datasets have been improved or added to the analysis since WMO 2014 would be beneficial in Section 2.

*Done. See our response to the first general comment of both reviewers.*

Line 13: For people with less familiarity with these stratospheric trend studies, it might not be clear why there is a focus on records starting before 1990 given that your analysis looks at the 2000-2016 period.

*Good point. We have added an explaining sentence at the beginning, have reworded the paragraph slightly, and have added some references.*

Lines 23-26: This is confusing if one is unaware that there were two SAGE instruments.

*Added a few words for clarification.*

Line 29: Improved in what way?

*By correcting for drift in instrument pointing. Explanation has been added.*

Lines 1-3: A brief mention of the different assumptions used in SWOOSH and GOZCARDS would be helpful to a reader trying to understand how independent these datasets are.

*SWOOSH and GOZCARDS ozone are largely constructed from the same data and in the same way, using the same assumptions. Explaining sentences have been added, and the text has been corrected to reflect the new versions of GOZCARDS (no use of ACE-FTS anymore) and SWOOSH that are actually used.*

Lines 17-28: The role of the ground-based data in this study is not quite clear to me. As far as I understand, they are not used in the average trend analysis, but they are also not used to quantitatively evaluate individual satellite records. It would be helpful if the authors could provide the rationale for their inclusion.

*Ground-based data provide an independent verification of the satellite results. We think they are important and we have added a short explanation at the beginning of the paragraph (and a reference), and also later in the discussion of the trend profiles.*

Lines 25-27: Is the fact that they are coherent over a wide range of altitudes and latitudes based on high vertical resolution profiles from satellites? Or on models? A reference would be useful here, since Harris et al. (2015) did not use ground-based profiles because of concerns about representativeness and the "coherency over latitude" would seem to negate that issue.

*This is a good point. Since the statement is not really important, since we did not find a good reference and to avoid unnecessary discussion, we decided to simply remove the sentence.*

Lines 29-35 and Page 5, lines 1-3: These drifts, while not statistically significant, are of the same order of magnitude, if not larger than, the trends reported here. It is clear from the abstract that a more detailed analysis of these drifts is being undertaken by LOTUS, but a somewhat deeper discussion of their relevance to results presented here is needed, particularly given that the differences between Harris et al. and these results seems to stem in part from lower drift in the records used here.

*This point has been addressed by adding some discussion, here and in other places, and by adding a table comparing the different data sets use.*

Lines 5-6: What is the rationale for using the 1998-2008 climatology for normalization?

*Added text to explain the rationale.*

Lines 14-16: It seems this point could be made more clearly by referring to sparser spatial and temporal sampling rather than "sparser sampling" meaning temporal and "geophysical differences" referring to the spatial sampling.

*OK, reworded.*

Figure 1: Presumably the grey line in Figure 1 refers the multi-model mean of the CCMVal2 models? This point should be clear in the caption and in the text in Section 2, and the authors should consider providing the full envelope of the models, as the range is fairly large.

*Added information to caption and text. We have also changed Figure 1 and are now plotting the envelope of the multi-model simulated anomalies.*

Line 27: Is there something missing here between "solar cycle" and "Reisel"?

*Thanks. Fixed.*

Section 3: It is clear from Figure 1 that there are data gaps in at least some of the ground-based records. How are these handled in the trend analysis?

*Only the available data can be used in the trend analysis. There was no special treatment for data gaps. Data gaps will, however, add some variability to the trend results. Since we did not do anything specific, and to avoided confusing details, we have not changed the text. However, the caption of Fig. 1 now mentions the gaps in some data records.*

Lines 18-25: I found this explanation confusing. It is unclear to me from this description how the first 2 regression terms are used. This seems to imply that only the last 4 terms are used and then a linear trend for 2000-2016 is fitted to the remaining residuals – if so then why are the first 2 regression terms included at all?

*Thanks for this important comment. The old text is indeed unclear and confusing. It has been reworded.*

Lines 20-23: On what years of data was the initial regression step performed?

*All years of the entire time series. Added text to clarify.*

Lines 2-3: The authors might want to refer to the Tegtmeier et al. paper on the SPARC Data Initiative ozone climatologies here.

*Thanks. Done.*

Figure 3: What uncertainty was used for the CCMVal 2 results? Is it based on the model range for all of the models or just on the ensemble mean? Please specify.

*See response to comment #11 by reviewer 1, and also to our response to reviewer 2's comment about Figure 1 above. Explaining sentences have been added to text.*

Line 3: Why was SBUV only used above 40 hPa?

*The lowest SBUV layers have very wide averaging kernel and include stratospheric and tropospheric contributions. True stratospheric profile information is only available for layers with layer centers above 40 hPa. Short explanation has been added.*

Figure 5: The caption needs several clarifications. It states that "uncertainty bars and yellow shading" give the +/- 2 x sigma values for all individual trends and seems to refer to the datasets used here (though it is unclear how both the uncertainty bars and yellow shading show the uncertainties for a single dataset), but then states that the yellow lines and shading show results from Harris et al. The WMO trend is apparently shown, but the color is not specified – is it the blue line? Finally, the clarification is again required for the model simulations – is this the ensemble mean? How are the uncertainties derived? I think perhaps things could be clarified if the sentence about uncertainty bars and shading were moved later in the paragraph.

*Thanks. The caption has been reorganized, and missing information has been added.*

Lines 20-21: Strictly speaking, a drift analysis requires comparison to independent datasets. It is unclear whether the authors are saying here that such an analysis has been performed and that the drift in the upper stratosphere has been determined to be 1-2% rather than the 6% used in Harris et al., or whether they are simply relying on the J-distribution analysis to argue for a small drift estimate. It is also unclear how this estimate relates to the estimates provide in Section 2, bottom of page 4, which describe drift estimates of 2-5% for the individual satellite records that make up the merged datasets.

*We agree. There was no real drift analysis. However, the spread of the trends in Fig. 4, and the (joint distribution) trend uncertainty in Fig. 5 indicate that some of the larger drift estimates of the past (e.g. 6% per decade) no longer apply. Text has been reworded (as has been the text in Section 2, bottom of page 4).*

Lines 16-21: For completeness, a brief discussion of the attribution of trends should be provided here.

*This point is also mentioned by reviewer #1 (his point 1.). It has been addressed by adding a few sentences.*

---

## Author Comment (AC2) · 11 Aug 2017

We thank both reviewers for their detailed and helpful comments. Please see the supplement to our response to reviewer 2 for all our detailed responses to both reviewers.